# COVID-19 in Older Patients: Assessment of Post-COVID-19 Sarcopenia

**DOI:** 10.3390/biomedicines11030733

**Published:** 2023-02-28

**Authors:** Almudena López-Sampalo, Lidia Cobos-Palacios, Alberto Vilches-Pérez, Jaime Sanz-Cánovas, Antonio Vargas-Candela, Juan José Mancebo-Sevilla, Halbert Hernández-Negrín, Ricardo Gómez-Huelgas, María Rosa Bernal-López

**Affiliations:** 1Internal Medicine Department, Regional University Hospital of Málaga, Instituto de Investigación Biomédica de Málaga (IBIMA-Plataforma Bionand), University of Málaga, Avda. Hospital Civil s/n, 29009 Málaga, Spain; 2Endocrinology and Nutrition Department, Virgen de la Victoria University Hospital, Instituto de Investigación Biomédica de Málaga (IBIMA-Plataforma Bionand), 29010 Málaga, Spain; 3Ciber Fisiopatología de la Obesidad y la Nutrición, Instituto de Salud Carlos III, 28029 Madrid, Spain

**Keywords:** COVID-19, sarcopenia, muscle strength

## Abstract

(1) Background: Acute COVID-19 infections produce alterations in the skeletal muscle, leading to acute sarcopenia, but the medium- and long-term consequences are still unknown. The aim of this study was to evaluate: (1) body composition; (2) muscle strength and the prevalence of sarcopenia; and (3) the relationship between muscle strength with symptomatic and functional evolution in older patients affected by/recovered from COVID-19; (2) Methods: A prospective, longitudinal study of patients aged ≥65 years who had suffered from COVID-19 infection between 1 March and 31 May 2020, as confirmed by PCR or subsequent seroconversion. Persistent symptoms, as well as anthropometric, clinical, and analytical characteristics, were analyzed at 3 and 12 months after infection. The degree of sarcopenia was determined by dynamometry and with SARC-F; (3) Results: 106 participants, aged 76.8 ± 7 years, were included. At 3 months postinfection, a high percentage of sarcopenic patients was found, especially among women and in those with hospitalization. At 12 months postinfection, this percentage had decreased, coinciding with a functional and symptomatic recovery, and the normalization of inflammatory parameters, especially interleukin-6 (4.7 ± 11.6 pg/mL vs. 1.5 ± 2.4 pg/mL, *p* < 0.05). The improvement in muscle strength was accompanied by significant weight gain (71.9 ± 12.1 kg vs. 74.7 ± 12.7 kg, *p* < 0.001), but not by an increase in lean mass (49.6 ± 10 vs. 49.9 ± 10, p 0.29); (4) Conclusions: Older COVID-19 survivors presented a functional, clinical, and muscular recovery 12 months postinfection. Even so, it is necessary to carry out comprehensive follow-ups and assessments that include aspects of nutrition and physical activity.

## 1. Introduction

The outbreak of coronavirus disease 2019 (COVID-19), caused by severe acute respiratory syndrome coronavirus-2 (SARS-CoV-2), has spread rapidly around the world, and it has had a huge impact on healthcare systems. The disease is associated with a wide spectrum of presentations, from mild, asymptomatic disease, to severe acute respiratory failure, resulting in damage to organs, such as myocardial dysfunction, hepatic injury, and renal injury.

Sarcopenia is a condition characterized by a progressive loss of muscle mass and strength. It arises as a consequence of aging; thus, it was originally confined to the elderly population. However, emerging evidence suggests that sarcopenia can develop at any age. Other than aging, possible causes, including nutrition, inflammation, or levels of vitamin D, have been recognized as potential mechanisms for the development of this disease.

Older people comprise the category most affected by COVID-19 infection, with almost 3 million cases in people aged ≥60 having been confirmed in Spain [1]. Infection produces a broad clinical spectrum of symptoms that are associated with lower nutrient intake, increased energy expenditure, or decreased nutrient absorption. As a consequence, nutritional requirements are often not met, resulting in weight and muscle loss. During the acute infection, patients are at risk of losing between 5 and 10% of their body weight [2,3]. In addition, systemic inflammation and reduced physical activity also contribute to this muscle wasting.

Sarcopenia is associated with limitations in physical function and quality of life [4,5], an increased risk of falls [6], vulnerability [7], and mortality [8]. Sarcopenia can occur acutely, over the course of days [9], or insidiously, over the course of months or years [8]. The European Working Group on Sarcopenia in Older People (EWGSOP) defines acute sarcopenia as incidental sarcopenia occurring within 6 months after stressful events, and it occurs most often in hospitalized patients [4]. The prevalence of sarcopenia is up to 15% in healthy older adults, and it can reach a rate of 69% in hospitalized patients. The prevalence of sarcopenia in individuals aged 60–70 years olds is reported as 15%, while the prevalence ranges from 11 to 50% in people >80 years old. [10]. Sarcopenia is likely when low muscle strength is detected, and diagnosis is confirmed by the presence of low muscle quantity or quality. A wide variety of tests and tools are available for the characterization of sarcopenia: the SARC-F questionnaire, grip strength, impedance, muscle ultrasound, gait speed, etc. Refs. [11,12] The selection of tools may depend on the patient, access to technical resources in the healthcare testing setting, or the purpose of the test.

The relationship between sarcopenia and COVID-19 has received substantial interest in the current literature. Sarcopenia can greatly affect the hospital prognosis of patients, as well as vulnerability to functional and physical deterioration after infection [13]. Published studies show that COVID-19 survivors are at an increased risk of sarcopenia during the weeks following infection [14,15], but few studies have evaluated long-term muscle strength. A study of patients who had recovered from COVID-19 showed a decrease in muscle strength and functionality, obtaining strength values below normal at the biceps brachii and the quadriceps femoris levels [16].

In this study with older patients affected by/recovered from COVID-19, the objectives were to evaluate: (1) body composition; (2) muscle strength and the prevalence of probable sarcopenia; and (3) the relationship between muscle force, disease severity, and symptomatic and functional evolution.

## 2. Materials and Methods

### 2.1. Study Design and Recruited Population

We conducted a prospective, longitudinal study of patients aged ≥65 years who were infected by SARS-CoV-2 between 1 March and 31 May 2020, as confirmed by RT-PCR or subsequent seroconversion using an antibody serological test. The participants were admitted to the Regional University Hospital of Malaga. The study did not include patients who died after admission or during follow-up, or those who had difficulty participating. Once potential participants were selected, the researchers contacted them to inform them about the study and invited them to participate. This study was approved by the Provincial Research Ethics Committee of Malaga (Spain), with the approval code “COVID-19/ANC”, dated 4 June 2021.

### 2.2. Written Informed Consent

Patients were invited to join the COVID-19 follow-up consultation group at the Internal Medicine Service of the Regional University Hospital Malaga, where they were informed about the study, and where they were required to sign the written informed consent. In the case that they were unable to sign the form, a legal guardian was asked to do so.

### 2.3. Clinical Data Collection

All participants gave their consent to collect their personal clinical data. Information regarding age, sex, comorbidities (hypertension, diabetes mellitus, obesity with a body mass index (BMI) ≥ 30, coronary heart disease, nonemboligenic ischemic stroke, peripheral arterial disease, liver disease, kidney disease, neoplastic disease, chronic obstructive pulmonary disease (COPD) or asthma), the need for hospital admission during acute illness, the length of hospital stay, and the need for admission to the ICU was collected. The Barthel Index is a widely validated item used to establish an individual’s functional condition [17]. This index refers to the ability to perform the basic activities of daily living. The Barthel Index is scored from 0 to 100, and it establishes the following categories: score <20: total dependence; 21–60: severe dependence; 61–90: moderate dependence; 91–99: mild dependence, and 100: independent. The comorbidity burden of patients was established with the age-adjusted Charlson comorbidity index [18], which is a system widely used for assessing life expectancy at 10 years, depending on the age at which it is assessed, and the subject’s comorbidities. In general, scores of 0–1 points indicate no comorbidity; scores of 2 points indicate low comorbidity; and scores of >3 indicate high comorbidity. Cognitive function was evaluated using the Mini-Mental State Examination (MMSE) [19]. The MMSE is the most-used test for assessing a wide range of domains, including attention, language, memory, orientation, and visuospatial competence. The maximum score that can be obtained is 30 points, with the following interpretation: no cognitive impairment (27–30 points); possible or borderline cognitive impairment (25–26 points); mild-to-moderate cognitive impairment (10–24 points); moderate-to-severe cognitive impairment (6–9 points); and severe cognitive impairment (<6 points). Finally, the FRAIL scale was used to detect frailty [20]; it evaluates five items: fatigue, resistance, aerobic capacity, illnesses, and weight loss. A score ≥3 indicates frailty.

### 2.4. Visits

Medical and nursing visits were made at 3 and 12 months after acute infection.

The clinicians carried out a general assessment of patients with an adequate history and physical examination, and they questioned patients about possible symptoms after acute infection.

Patients underwent a first visit and a follow-up visit with a nurse, during which weight, height, BMI, waist circumference (WC), hip circumference, heart rate, and blood pressure measurements were obtained. Weight, lean mass quantification, and fat mass were assessed using bioelectrical impedance (BIA) via an electronic scale (Tanita Body Composition Analyzer (TBF-300 MA) Tanita Corporation, 1–14–2 Maeno-cho, Itabashi-ku, Tokyo, Japan). The Tanita system is an easy and effective tool for measuring body composition. It has a high level of test-pretest validity although it presents a somewhat higher margin of error as compared to the gold-standard body composition tests (hydrodensitometry and dual-energy X-ray absorptiometry (DEXA) tests). The measurement of height was performed with participants shoeless and a wall stadiometer (Stadiometer Barys Electra Model 511-300-A0A, ASIMED, Tokyo, Japan). BMI was measured by the division of weight (kg) by height squared (m^2^). The measurement of WC was carried out halfway between the last rib and the iliac crest, by means of an anthropometric tape. Blood pressure was obtained via an automated electronic sphygmomanometer (OMRON M7 (HEM-780-E), OMRON Healthcare Co., Ltd., Kyoto, Japan).

Blood samples were drawn after a 12-h fast, and biochemical measurements (hemoglobin, lymphocytes, D-dimer, gamma-glutamyl transferase (GGT), oxaloacetic glutamic transaminase (GOT), lactate dehydrogenase (LDH), C-reactive protein (CRP), ferritin, and interleukin-6) were determined using routine methods at the Clinical Analysis Laboratory of the Regional University Hospital of Málaga.

### 2.5. Sarcopenia Assessment

The diagnostic criteria for sarcopenia were based on those of the 2018 European Working Group on Sarcopenia in Older People (EWGSOP-2). The strength of the hand was assessed by dynamometry (Jamar Plus) using the Southampton protocol [21]. This protocol consists in measuring grip strength with the subject seated in a chair with the forearms resting on the arms of the chair. The wrist is placed in a neutral position with the thumb on the top of the chair and three measurements are taken on each side, alternating sides, and starting with the right hand. The best result from each side is selected. Probable sarcopenia was defined as values of <16 kg in women and <27 kg in men [4].

In addition, all participants completed the SARC-F questionnaire, which is a self-screening questionnaire for sarcopenia risk, and it allows for the identification of potential cases [22]. It evaluates five components: strength, help walking, getting up from a chair, climbing stairs, and falls. Scores range from 0 to 10 points. A value ≥ 4 points is an indication of sarcopenia.

Sarcopenia grades were evaluated in the total population, by sex, and with the consideration as to whether a patient needed to be hospitalized, or not, during the acute phase of infection.

### 2.6. Dietary Assessment

All participants completed a validated food frequency questionnaire of 14 items [23] to assess their adherence to the Mediterranean diet. The questionnaire categorizes responses as high (12–14 points); moderate (8–11 points); low (5–7 points); or very low adherence (<5 points).

### 2.7. Statistical Analysis

Data analysis was performed with the IBM SPSS v22.0 program. Quantitative variables with a normal distribution were expressed as means ± standard deviation (SD), and qualitative variables were expressed as percentages. Student’s *t*-test was used to compare quantitative variables, and the chi-squared test was used to compare qualitative variables. Pearson’s correlation coefficient was used to determine the correlations between quantitative variables.

## 3. Results

A total of 150 individuals were identified as possible participants. Out of those, 39 refused to participate, and 111 attended the first visit. Five patients were excluded from the study. Finally, 106 participants of both sexes (51.9% men and 48.1% women) were included. From them, 80 participants required hospital admission during the acute phase, and 26 patients did not (see Figure 1).

The clinical and epidemiological variables of the study population at diagnosis of SARS-CoV-2 infection are shown in Table 1. The mean age of all participants was 76.8 ± 7 years. The age of females was slightly higher than that of males (76.5 ± 6.5 years for males and 77.3 ± 7.2 years for females). At the functional level, prior to infection, most participants were independent or had a mild level of dependence for the performance of the basic activities of daily living, with males having a higher degree of independence than females. In this sense, a worsening was observed after 3 months in the total population, as compared to baseline conditions (87.9 ± 21.3 points vs. 93.7 ± 15.7 points), and in both sexes (88.1 ± 19.2 vs. 95.5 ± 12.6 points in men and 87.8 ± 23.2 vs. 91.7 ± 18.4 points in women). However, 12 months after infection, an improvement was observed, achieving almost baseline conditions (total population: 93.6 ± 15.2 points; men: 94.6 ± 15.2; and women: 92.1 ± 16). No significant differences were found between the sexes in the MMSE scores or the FRAIL Index scores.

Our population displayed a high comorbidity grade (4.6 ± 1.8 points), with it being higher in males than in females (4.9 ± 1.9 vs. 4.2 ± 1.6, respectively). The main comorbidities were arterial hypertension, dyslipidemia, heart disease, and diabetes mellitus. Significant differences in the presence of cardiopathy, smoking, COPD, neoplasms, SAHS, and alcoholism, depending on sex, were found, with those being most prevalent among the men. However, females showed a higher rate in the incidence of psychological disorders such as depression and anxiety.

The symptomatic evolution of all participants, as assessed at the follow-ups, is shown in Table 2. Significant differences were found in the most frequent symptoms, including asthenia (*p* < 0.001), dyspnea (*p* = 0.003), and weight loss (*p* < 0.001), between the two follow-ups. At 3 months postinfection, 61.7% of the patients reported two or more persisting symptoms, while at 12 months, this percentage had dropped to 8.1% (*p* < 0.001). At 12 months, all analyzed symptoms presented as improving although the rate of recovery was lower in the female sex.

The anthropometric, body composition, and analytical parameters throughout the follow-up are shown in Table 3. After 12 months, there was a significant increase in weight, globally (+2.8 kg) and in both sexes (+2.9 kg in males and +1.4 kg in women). Body fat percentage and fat mass increased significantly globally (+1.8% and +1.2 kg, respectively) and in males (+2.2% and +1.7%, respectively) although it did not reach a level of significance (+0.2% and +0.5 kg, respectively) in women. However, no significant differences in lean mass were found. After 12 months, significant decreases in neutrophil count (−1590 U/μL), D-dimer (−242.5 ng/mL), and interleukin-6 (−3.2 pg/mL) were observed, as well as an increment in the lymphocyte count (+410 U/μL).

Table 4 shows the values determined via dynamometry and the SARC-F, as well as the percentage of patients diagnosed with sarcopenia according to the results, stratified by sex. An improvement in dynamometry values was observed globally (+1.2 kg), and by sex (males +2.5 kg and women +0.5 kg) although the improvement was only significant in males. The SARC-F scores were also better at 12 months, but only significant differences were present in males. The percentage of patients diagnosed with probable sarcopenia was lower at 12 months, both globally and in both sexes, although these differences were significant only in men. Hospitalized patients had lower dynamometry values (−1.7 kg and −1.5 kg at 3 and 12 months, respectively) and higher SARC-F scores (+1.1 points and +1.0 point, at 3 and 12 months, respectively) than those of nonhospitalized patients. These differences were only significant at 3 months postinfection.

Correlations between muscle strength and epidemiological, anthropometric, clinical, and analytical factors were found globally. Positive correlations between muscle strength and lean mass (*r*: 0.68; *p* < 0.001), weight (*r*: 0.5; *p* < 0.001), independence (*r*: 0.35; *p* < 0.001), and cognitive level (*r*: 0.25; *p* < 0.001) were found. In contrast, negative correlations between muscle strength and SARC-F (*r*: −0.55; *p* < 0.001), frailty (*r*: −0.47; *p* < 0.001), age (*r*: −0.37; *p* < 0.001), comorbidities (*r*: −0.35; *p* < 0.001), an increase in symptoms post-infection (*r*: −0.22; *p* = 0.03), and lymphopenia (*r*: −0.22; *p* = 0.04) were found. No significant correlations with days of hospital admission or elevated IL-6 levels were found.

Finally, adherence to the Mediterranean diet was evaluated in all patients. No significant differences were found in any times of follow-up, with a moderate rate of adherence.

## 4. Discussion

With this study, we demonstrated that, in older patients (whether hospitalized, or not) functional, clinical, and muscular recovery was achieved 12 months after COVID-19 infection. This recovery at the muscular level translates to an increase in muscle strength and a lower degree of sarcopenia in both sexes although this improvement was only significant in males. These results were associated with greater values for weight, functional independence, and the absence of cognitive impairment. Muscle mass also increased slightly, but not significantly.

The identification of sarcopenia in our study was performed via the detection of muscle strength, as measured by dynamometry, using grip strength. Data recorded by dynamometry in our study reflect that, 3 months after infection, nearly 80% of the patients presented with low muscle strength and probable sarcopenia. This percentage is higher than that described in previous studies [24]. We identified a higher percentage of sarcopenia, or patients at risk of sarcopenia, among women, unlike most studies, where no association between sarcopenia and sex was found [8,25] or a predominance in males was found [26]. The greater incidence in males appears to be related to the higher degree of smoking [27], alcoholism [28], and diabetes [29]. These three factors also predominated in the males in our sample; therefore, the higher incidence of sarcopenia found in females could be due to a greater persistence in symptoms and a slower recovery after infection.

In addition, all participants completed the SARC-F self-assessment questionnaire for identifying at-risk patients. The SARC-F questionnaire is a simple and inexpensive method that allows us to detect individuals at risk of adverse sarcopenia outcomes [4], and it has been used in multiple studies, including ones conducted with elderly patients during the COVID-19 pandemic [30,31]. It is recommended as a screening method, and it is capable of detecting severe cases of sarcopenia, but it has a low-to-moderate level of sensitivity for predicting low muscle strength. [4]

Ageing produces changes in body composition [32], and sarcopenia increases with age [33], which could be another reason for the high rate of sarcopenia in our study, as previously demonstrated by other authors [34]. However, the degree of sarcopenia may be influenced by various risk factors, such as age, a prolonged hospital stay > 7 days, the need for invasive mechanical ventilation, and obesity [24]. Nonetheless, the specific risk factors for post-COVID sarcopenia are not yet well-known.

Hospitalization is associated with acute alterations in the sarcopenia status of patients [30], but, in addition, the presence of sarcopenia conditions is associated with complications during a hospital admission and with a higher rate of readmission [35]. In our study, at 3 months, subjects who had required hospitalization had a higher percentage of sarcopenia, which was not significant compared to nonhospitalized patients. However, both percentages are higher than those reported in the literature [24]. At 12 months, an improvement in both groups was found, but there were no significant differences between them.

A lack of exercise is another major risk factor for sarcopenia [36]. Immobilization results in significant changes in muscle cross-sectional area, volume, and mass, which promotes metabolic dysfunction and leads to impaired functionality [37]. Levels of physical inactivity were higher during the COVID-19 lockdowns, which forced the elderly population to stay at home, depriving them of regular physical activity, which accelerates the loss of strength and muscle function [38]. This immobilization probably explains the high degree of sarcopenia found in nonhospitalized patients. This increased social isolation has prevented the elderly from participating in their group activity programs, greatly affecting their physical condition [38], as has occurred due to previous pandemics.

There is increasing evidence that hyperinflammation is closely related to the development of sarcopenia. Elevated C-reactive protein (CRP), IL-6, and TNF-alpha have shown the strongest correlation with sarcopenia and frailty, resulting in extreme muscle wasting due to the promotion of catabolic signals mediated through these proinflammatory cytokines [39]. A direct relationship between increased IL-6 [40] and TNF- alpha concentrations [41] is associated with decreases in muscle mass and strength muscle. In our study, IL-6 levels after 12 months displayed a significant decrement although this decline was not correlated with increases in muscle strength.

As previously mentioned, our data reflect an improvement in muscle strength at 12 months after infection, but they do not correlate significantly with gains in lean mass. Sarcopenia was once synonymous with a loss of muscle mass, but nowadays, other parameters such as loss of muscle strength and physical performance are considered to be more important. The relationship between low muscle mass and disability is not well-defined, but there is a clear association with low muscle strength [42]. The decline in muscle strength is known to occur much more quickly than the concomitant loss of muscle mass [43], and in this sense, some authors prefer the term “dynapenia” to refer to this loss of muscle strength with age [44].

The measurement of body composition in our study was performed via bioelectrical impedance analysis (BIA) using the Tanita system. BIA is based on the principle that the conductivity of water in the body varies in the different compartments, thus allowing differentiation between fat mass and lean mass, and it is a simple and widely used method. Its main disadvantage compared to the gold-standard tests for body composition (computerized axial tomography (CT), magnetic resonance imaging (MRI), and dual-ray X-ray absorptiometry (DEXA), is its greater margin of error and its limitations in patients with hydrological decompensation.

In addition, a low-protein diet is considered to be another factor that accelerates the onset of sarcopenia in older adults [45]. A high prevalence of malnutrition has been found among older COVID-19 survivors [46]. Therefore, it is essential to routinely perform nutritional studies with these patients [46]. Patients with sarcopenia detected after infection may benefit from oral nutritional supplementation (ONS) providing at least 400 kilocalories (kcal) per day, with 30 g of protein or more. Such a strategy should be continued for at least 30 days [47] although we propose that, in older patients with post-COVID sarcopenia, this oral supplementation could be maintained for at least 3 months post-acute infection.

Not only protein deficiency is responsible for sarcopenia. An inadequate intake of other micronutrients can contribute to it. High adherence to the Mediterranean diet is associated with a lower incidence of chronic diseases and less physical deterioration in old age [48], which may reduce the risk of experiencing sarcopenia among the elderly [49]. In this sense, our population exhibited a moderate level of adherence to the Mediterranean diet. Another important component for improving muscle strength and reducing the risk of sarcopenia is physical activity. There are multiple, validated guidelines for improving physical activity that could help in the recovery of patients who have suffered COVID-19 infection. [50]. It is recommended that older people admitted to the hospital or in home-based quarantine for COVID-19 should undergo regular physical training [51]. The practice of low-intensity aerobic with resistance training produces an improvement in hand grip strength in older adults with post-COVID sarcopenia [52]. Home exercise programs have been shown to improve different components of health and fitness, such as muscular strength, muscular endurance, and balance, in older people [53]. Thus, in times of restricted physical activity due to pandemic situations, home-based resistance exercises are an alternative for preserving the physical fitness of older adults. A minimal form of exercise supervision is recommended, e.g., through weekly visits and/or telephone calls.

Sarcopenia and frailty are related, and sometimes they overlap, especially if we focus on the physical aspect of Fried’s definition of frailty, which includes a low walking speed, a low grip strength, and weight loss [54]. This relationship is present in our study; greater frailty means less muscle strength and more sarcopenia. In addition, our data relate such a relationship between sarcopenia, functional dependence, and comorbidities, as noted in previous studies [55,56]

In the older population, sarcopenia has become an important focus of research and public policy debate. The loss of muscle strength contributes directly to exercise intolerance and the impairment of daily activities, making it a strong determinant of quality of life, mortality, and healthcare spending [57]. Despite its clinical importance, sarcopenia remains poorly understood and poorly managed in routine clinical practice. Our study emphasizes the importance of knowing how to recognize those older patients who are at risk, especially after overcoming a severe infectious process such as COVID-19, as well as the need to establish guidelines for follow-up and action through a specific and controlled assessment that allows for the identification and treatment of the effects of muscle damage caused by the infection.

## 5. Conclusions

COVID-19 infection has been associated with a wide spectrum of symptoms and an increased risk of sarcopenia during the weeks following infection, especially in elderly patients. Along with infection, the increase in physical inactivity during the pandemic has contributed to further physical deterioration and the loss of muscle strength. Our results show a progressive functional, clinical, and muscular recovery occurring 12 months after acute infection in older patients who survived the disease, with this recovery being greater in men. Even so, we found a high percentage of sarcopenia and a persistence of symptoms. Therefore, to ensure this recovery, it is necessary to perform structured and coordinated follow-ups, promoting healthy lifestyle habits, as well as to know the main sequelae caused by the infection.

## Figures and Tables

**Figure 1 biomedicines-11-00733-f001:**
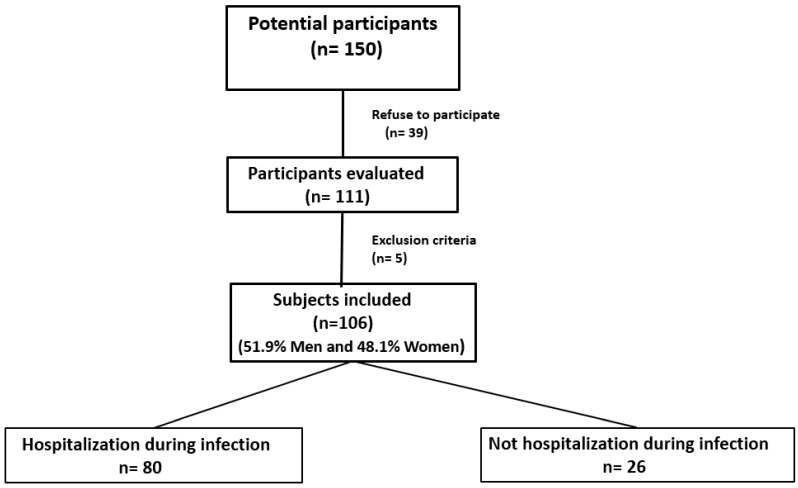
Flowchart diagram of the selection of patients for our study.

**Table 1 biomedicines-11-00733-t001:** Clinical and epidemiological characteristics of the total study population, and stratified by sex, at diagnosis of SARS-CoV-2 infection (baseline). (Variables are expressed as means ± standard deviation (SD) and *p*-values.).

	Total (*n* = 106)	Men (*n* = 55)	Women (*n* = 51)	*p*
Barthel Index (points)	93.7 ± 15.7	95.5 ± 12.6	91.7 ± 18.4	0.3
Charlson Index (points)	4.6 ± 1.8	4.9 ± 1.9	4.2 ± 1.6	0.3
MMSE Index (points)	28.05 ± 3.5	28.26 ± 2.96	27.8 ± 3.98	0.4
FRAIL Index (points)	1.34 ± 1.25	1.32 ± 1.32	1.37 ± 1.19	0.6
Hypertension (%)	81.1	78.2	84.3	0.2
Dyslipidemia (%)	47.2	40	54.9	0.1
Cardiopathy (%)	34.9	45.4	23.6	0.02
Diabetes (%)	29.3	32.7	25.5	0.5
Smoking (%)	28.3	47.3	7.8	<0.001
COPD (%)	16	25.5	5.9	<0.001
Depression (%)	15.1	1.8	31.4	<0.001
Dementia (%)	13.2	12.7	13.7	0.9
Neoplasm (%)	13.2	20	5.9	0.04
Anxiety (%)	13.2	3.6	23.5	<0.001
AF (%)	12.3	16.4	7.8	0.4
CKD (%)	11.3	12.7	9.8	0.7
SAHS (%)	7.5	12.7	2	0.06
Stroke (%)	6.6	9.1	3.9	0.4
Asthma (%)	6.6	5.5	7.8	0.7
Alcoholism (%)	4.7	9.1	0	0.05
Liver disease (%)	3.8	1.8	5.9	0.2
Hemodialysis (%)	1.9	0	3.9	0.2

MMSE: Mini-Mental State Examination; COPD: chronic obstructive pulmonary disease; AF: atrial fibrillation; CKD: chronic kidney disease; SAHS: sleep apnea/hypopnea syndrome.

**Table 2 biomedicines-11-00733-t002:** Symptomatic evolution, 3 and 12 months post-acute infection, in the total study population and as stratified by sex. (Variables are expressed as percentages; *p*-values of <0.05 are considered to be significant.)

Symptoms	Population	3 Months	12 Months	*p*
Asthenia (%)	Total	51.9	26	<0.001
	Men	41.8	10.3	0.003
	Women	62.7	42.1	0.01
Dyspnea (%)	Total	50.9	20.8	<0.001
	Men	49.1	7.7	<0.001
	Women	52.9	34.2	0.018
Weight loss (%)	Total	27.4	1.3	<0.001
	Men	38.2	2.6	0.001
	Women	15.7	0	
Cough (%)	Total	20.8	7.8	<0.001
	Men	20	5.1	0.02
	Women	21.6	10.5	0.01
Anxiety (%)	Total	17	3.9	0.02
	Men	21.8	1.3	0.08
	Women	11.8	7.9	0.01
MMII Weakness (%)	Total	17	6.5	0.02
	Men	14.5	5.1	0.10
	Women	19.6	7.9	0.08
Chest pain (%)	Total	13.2	3.9	0.06
	Men	14.5	5.1	0.18
	Women	11.8	2.6	0.18
Anosmia–Ageusia (%)	Total	12.3	0	
	Men	9.1	0	
	Women	15.7	0	
Mucus (%)	Total	9.4	0	
	Men	14.5	0	
	Women	3.9	0	
Diarrhea (%)	Total	7.5	0	
	Men	12.7	0	
	Women	2	0	
Headache (%)	Total	5.7	5.2	0.66
	Men	1.8	5.1	0.57
	Women	9.8	5.3	0.16
Skin lesions (%)	Total	4.7	0	
	Men	3.6	0	
	Women	5.9	0	

**Table 3 biomedicines-11-00733-t003:** Anthropometric, body composition, and analytical parameters at 3 and 12 months postinfection, in the total study population and as stratified by sex. (Variables are expressed as means ± standard deviation (SD) and *p*-values.)

		3 Months	12 Months	*p*
Weight (kg)	Total	71.9 ± 12.1	74.7 ± 12.7	<0.001
	Men	78.6 ± 9.9	81.5 ± 10.5	<0.001
	Women	66.6 ± 10.9	68 ± 11.3	0.03
BMI (kg/m^2^)	Total	28.1 ± 3.5	29.1 ± 4.5	<0.001
	Men	28.4 ± 3.4	29.5 ± 3.7	<0.001
	Women	28.3 ± 5.2	28.9 ± 5.3	0.01
Body fat (%)	Total	32.2 ± 7.3	33 ± 8.4	<0.05
	Men	28.5 ± 5.4	30.7 ± 6.8	0.001
	Women	35.7 ± 9	35.9 ± 9	0.25
Fat mass (kg)	Total	23.9 ± 7.4	25.1 ± 9.2	0.002
	Men	23.4 ± 6.4	25.1 ± 7.1	0.005
	Women	24.8 ± 10.8	25.3 ± 11.4	0.15
Lean mass (kg)	Total	49.6 ± 10	49.9 ± 10	0.29
	Men	56.5 ± 6.1	57.3 ± 7.2	0.18
	Women	41.7 ± 4.9	41.8 ± 4.6	0.92
Waist circumference (cm)	Total	96.9 ± 10.8	99.1 ± 15.9	0.02
	Men	101.3 ± 7.3	105.9 ± 15.3	0.04
	Women	90.3 ± 11.8	91.9 ± 13.9	0.17
Hb (g/dL)	Total	14.4 ± 14.1	14.4 ± 14.2	0.9
Platelets (U/µL)	Total	272.8 ± 113.9	257.1 ± 145.3	0.3
Leukocytes (U/µL)	Total	7700 ± 4356	9520 ± 5678	0.5
Neutrophils (U/µL)	Total	5800 ± 3021	4210 ± 2134	<0.0001
Lymphocytes (U/µL)	Total	1130 ± 596	1540 ± 578	<0.0001
TP (sg)	Total	92.6 ± 18.4	87.1 ± 15	0.7
INR	Total	1.13 ± 0.8	1.09 ± 0.6	0.6
D-dimer (ng/mL)	Total	700.5 ± 1199	458 ± 545	<0.05
LDH (U/L)	Total	168.8 ± 42.5	181.8 ± 41.8	0.3
GOT (U/L)	Total	29.3 ± 12.7	26.1 ± 24.1	0.3
GGT (U/L)	Total	32.8 ± 23.3	24.4 ± 13.3	0.07
Ferritin (ng/mL)	Total	93 ± 144	90 ± 110.9	0.7
CRP (mg/mL)	Total	4.5 ± 7	4.2 ± 6.8	0.6
IL-6 (pg/mL)	Total	4.7 ± 11.6	1.5 ± 2.4	<0.05

Hb: hemoglobin; TP: prothrombin time; INR: international normalized ratio; LDH: lactate dehydrogenase; GOT: glutamate oxaloacetate; GGT: gamma-glutamyl transferase; PCR: C-reactive protein; IL-6: interleukin-6.

**Table 4 biomedicines-11-00733-t004:** Dynamometry values, SARC-F scores, and percentages of patients diagnosed with sarcopenia according to both criteria, in the total study population and as stratified by sex. Variables are expressed as means ± standard deviation (SD) and percentages; *p*-values of <0.05 are considered to be significant.

		3 Months	12 Months	*p*
Dynamometry value (kg)	Total	16.8 ± 8.3	18. ± 8.7	0.06
	Men	21 ± 8	23.5 ± 8.3	0.03
	Women	12 ± 4.2	12.6 ± 4.7	0.45
Possible sarcopenia diagnosis by dynamometry (%)	Total	81.4	57.35	0.07
	Men	78	48	0.01
	Women	84.8	66.7	0.56
SARC-F value (points)	Total	2.6 ± 1.9	2 ± 1.8	0.06
	Men	2.23 ± 1.67	1.16 ± 1.2	0.01
	Women	3.11 ± 1.8	2.85 ± 1.92	0.31
Sarcopenia diagnosis by SARC-F (%)	Total	32.65	19	0.07
	Men	32.7	10.5	0.03
	Women	32.6	27.5	0.10

## Data Availability

Data cannot be shared publicly because of confidential data. Data are available from the IBIMA Institutional Data Access/Ethics Committee (contact via D. Andres Gonzalez Jimenez, ue.amibi@acitamrofnioib) for researchers who meet the criteria for access to confidential data.

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
