# Peer review of "COVID-19 in Older Patients: Assessment of Post-COVID-19 Sarcopenia"

_biomedicines, 2023, doi:10.3390/biomedicines11030733_

Round 1

Reviewer 1 Report

SUMMARY

In this manuscript, López-Sampalo et al. examined the incidence of and putative recovery from COVID-associated muscle wasting in elderly. Overall, major revisions are required prior to publication and those concerns are listed below.

MAJOR COMMENTS

METHODS  

  1. EWGSOP2 clearly presents an algorithm for case finding, making a diagnosis, and quantifying severity of sarcopenia in practice. The SARC-F is a self-screening questionnaire for sarcopenia risk and to find potential cases, and it is NOT a diagnostic test for sarcopenia. Please change this in methods and discussion.
    1. Ideally for diagnosis, Lopez-Sampalo et al. should have used three different laboratory-assessed criteria for sarcopenia diagnosis and severity thereof (i.e. muscle strength, function, and mass (see below)), each of which have published EWGSOP2 standards, after they administered the SARC-F.  

                                                    i.     Please provide a discussion in your paper regarding to the drawbacks of using SARF-F for sarcopenia diagnosis, even when combined with a validated test such as grip strength. You may include use the references Malmstrom et al. (2016) and Ishi et al (2014) to support the use of SARC-F, but ultimately it has low to moderate strength to even predict low muscle strength.

  1. The original EWGSOP and EWGSOP2 presented the most preferred measurements for sarcopenia diagnosis. Please provide a general statement on why only the grip strength was used.
    1. Muscle strength: Grip strength (Jamar) AND lower body strength (leg extension using Cybex or weight machines).

                                                    i.     First,in your methods, please specify exactly how the grip strength test was performed, for example, which protocol was used? Did you use the Southampton protocol or another protocol?

                                                  ii.     Remember that lower body strength should always be assessed in conjunction with grip strength since it is more relevant to the mobility of patients. Please provide a statement in the discussion  of the importance of assessing lower body strength for mobility in sarcopenia cases. Francis P & Cormack W  (2017) is a good reference to start with.   

    1. Muscle performance/function: The Short Physical Performance Battery (SPPB) OR 6-M Gait Speed, Timed Up and Go (TUG), and 4-Step Stair Climb performance. Please provide a statement the reasons why you did not test performance and function of the patients. These would have been easily done if you can do the grip strength. All you need is a chair, a measure tape and/or a timer, for example!
    2. Muscle mass: ASM or ASM/H2 from DEXA scans. Please provide a statement on why BIA was the preferred method for determining muscle mass in this study.

                                                    i.     Obviously, the TANITA system (BIA) has major drawbacks when it comes to assessing body composition and this needs to be addressed in the discussion. Please provide a discussion on correlation of the TANITA System with gold standard methods for body composition, including MRI, DEXA and hydrostatic weighting.

                                                  ii.     Also, you need to include the test-retest validity for the TANITA system in the methods to prove consistency/validity of your results.

  1. Please provide an explanation why you did not assess PA and levels directly in these patients; for example, by an activity questionnaire. Exercise and nutrition are the two biggest regulators, beyond aging and disease, of muscle mass and function.  
  2. Please provide a description in the methods what each indices measures, validity of tests, and how they are administered etc., including Barthel, Charlson, MMSE, and Frail.

 RESULTS  

  1. Please make a CONSORT flow diagram of recruitment flow (or equivalent). This will make the recruitment process and inclusions/exclusions easier to follow.
  2. If you happen to have the baseline values as well (i.e. 0, 3 and 12 months), include them as well for all tables. It would strengthen the paper significantly.
    1. Also, please correct the spelling (weight loss; circumference; possible) and make sure you use to correct English language for outcomes. For example, “Dimero D” should be “D Dimer” and “Ferritine” should be “Ferritin” etc.
    2. Check that all p-values are correctly written. INR seems incorrect.
  3. Table 1: It is unclear at what time point these measures were obtained. Make it clear in the sub-figure. Why not include all time points if you have them for clarity? Also make it clear which indices you are referring to in the written results section.  

 DISCUSSION

  1. You are not showing increased muscle mass or lean mass at 12 months. You must remove this from the first paragraph in the discussion. You merely show increased grip strength in males, not even females. Address and reword please.
  2. Please discuss whether the SARC-F and grip strength results can be explained by just general fatigue/malaise from COVID-19 at 3 months and not muscle loss/deterioration per se. If you would have had baseline values, you would have been able to prove muscle/strength loss, but you have not shown this in this manuscript.
  3. In addition to biological aging, sedentary living/lack of exercise is the biggest determinant of age-related muscle wasting.
    1. Please discuss the role of exercise & lack thereof in more detail, specifically resistance exercise & weightbearing exercises, and why it is important for mitigating age-related sarcopenia. Also, expand on how difficult it was and is during pandemics for elderly to partake in traditional exercise programming and that home-based resistance exercise (HBRE) will be key in the future for healthy aging and mitigation of muscle loss.  
  1. Similarly to exercise, please include a section on enhanced protein-intake & supplementation and sarcopenia mitigation, especially in COVID19 patients. Also, what is most important for COVID patients around the world clinically? A higher intake of protein and a mixed/healthy diet overall? Or consuming the Mediterranean diet, specifically? Please make this clear in the discussion.
  2. What would be the optimal treatment for mitigating sarcopenia in COVID 19 patients?  Specifically, please include a section on protein/multi-ingredient supplementation, resistance exercise, and home-based resistance exercise. Are there clinical trials on this? After all, we want clinical solutions.  

MINOR COMMENTS

  1. Please correct the punctuation, spaces between words, and spelling in the paper.
  2. Include institutional review board number and acceptance date in the methods section.

Reviewer 2 Report

-Abstract: What is design of study?

-Introduction: What is hypothesis of study?

-Methods: What is method to quantify fat mass and lean mass?

-Results: Table 5 may be removed.

-Dicussion: More studies using the SARC must be mentioned and discussed: please see: 

  • PMID: 34545924 PMCID: PMC8322632 DOI: 10.1007/s12603-021-1663-x
    • PMID: 35744091
    •  
    • PMCID: PMC9231342 
    • DOI: 10.3390/medicina58060828

Reviewer 3 Report

This is an interesting manuscript about the correlation between COVID-19 and the sarcopenia.

The aims of this study was to demonstrated a relation between elderly patient (hospitalized ot not) and a functional clinical muscular recovery clinical and muscular recovery achieved after 12 months of infection especially in male sex. Even so, it is necessary to carry out a comprehensive follow-up and assessment, including a nutritional and physical activity approach.

The manuscript is well written and after some major revisions could be suitable for publication.

Although the overall conclusion reached by the authors is reasonable, as currently presented, this manuscript has a number of deficiencies that need to be addressed, outlined below: There are numerous errors of “Tense and Grammar” throughout the manuscript. Therefore, a diligent editing is in order to fix the ENGLISH language.

Minor comments

Abstract

Line 5: Please consider writing “elderly” instead of the present sentence.

Introduction

I suggest to start the section like this

The outbreak of coronavirus disease 2019 (COVID-19), caused by the severe acute respiratory syndrome cornonavirus-2 (SARS-CoV-2), has spread rapidly around the world and it has a huge impact on healthcare systems. The disease is associated with a wide spectrum of presentations, from mild asymptomatic disease to severe acute respiratory failure, resulting in damage to organs such as myocardial dysfunction, hepatic injury, and renal injury.

Sarcopenia is a condition characterized by a progressive loss of muscle mass and strength. It arises as a consequence of aging so it was originally confined to the elderly people. However, emerging evidence suggests that sarcopenia can develop at any age. Other than aging, possible causes, including nutrition, inflammation, vitamin D, are recognized as potential mechanisms to developmentof this disease.

Line 1-2: Please consider writing “Elderly people is the category most affected by” instead of the present sentence.

Line 3-4: Please consider delete this sentence.

Line 11-12: Please delete this sentence.

Line 13: Please consider writing “sarcopenia can occur” instead of the present sentence.

Line 17-20: Please consider writing “The prevelence of the Sarcopenia is up to 15% in healthy older adults  and can reach  a rate of 69% in ospedalized patients. Sarcopenia prevalence in 60–70-year-olds is reported as 15%, while the prevalence ranges from 11 to 50% in people >80 years.” instead of the present sentence.

Line 20-21: Please delete this sentence.

Line 26: Please consider writing “The relationship between sarcopenia and COVID-19 has received substantial interest in the current literature.” instead of the present sentence.

Line 35: Please consider writing “Elderly” instead of the present sentence.

Materials and Methods

2.1

Line 1-2: Please consider writing “who was infected by SARS-CoV-2” instead of the present sentence.

Line 1-2: Please explain what do you mean for subsequent seroconversion, did you test them using antibody serological test? Please consider writing “RT-PCR” instead of the present sentence.

Line 4-6 Please consider rewriting this sentence.

2.2

Line 1-2: Please consider writing “Patients were invited to join the COVI-19 follow up consultation group at the Internal Medicine Service of the Regional University Hospital Malaga where they are informed about the study and where they have to sign the sign the written informed consent” instead of the present sentence.

Line 4: Please consider writing “are not able to” instead of the present sentence.

Results

Line 3: Please consider writing “Five patients were ecìxcluded from the study.” instead of the present sentence.

Line 5-6: Please explain better this sentence.

Line 7-8: not understand what do you mean for dependence.

Line 20-21: Please consider writing “show a higher rate of incidence of” instead of the present sentence.

Line 44-47 Please consider rewriting this sentence.

Discussion

Line 24-26 Please consider rewriting this sentence.

Line 56-58 Please consider rewriting this sentence.

Line 65-67 Please consider rewriting this sentence.

Conclusions

Not clear the conclusions. Please improve this section.

Round 2

Reviewer 1 Report

Thanks for adding the changes I asked for - it tempered your manuscript, and it is a now publishable. Best of luck. 
